# Empirical Comparison of Imputation Methods for Multivariate Missing Data in Public Health

**DOI:** 10.3390/ijerph20021524

**Published:** 2023-01-14

**Authors:** Steven Pan, Sixia Chen

**Affiliations:** Department of Biostatistics and Epidemiology, University of Oklahoma Health Sciences Center, 801 NE 13th St, Oklahoma City, OK 73104, USA

**Keywords:** imputation, multivariate missingness, nonresponse bias, public health data

## Abstract

Sample estimates derived from data with missing values may be unreliable and may negatively impact the inferences that researchers make about the underlying population due to nonresponse bias. As a result, imputation is often preferred to listwise deletion in handling multivariate missing data. In this study, we compared three popular imputation methods: sequential multiple imputation, fractional hot-deck imputation, and generalized efficient regression-based imputation with latent processes for handling multivariate missingness under different missing patterns by conducting descriptive and regression analyses on the imputed data and seeing how the estimates differ from those generated from the full sample. Limited Monte Carlo simulation results by using the National Health Nutrition and Examination Survey and Behavioral Risk Factor Surveillance System are presented to demonstrate the effect of each imputation method on reducing bias and increasing efficiency for the parameter estimate of interest for that particular incomplete variable. Although these three methods did not always outperform listwise deletion in our simulated missing patterns, they improved many descriptive and regression estimates when used to impute all incomplete variables at once.

## 1. Introduction

Missing data is a common issue for many researchers when conducting statistical analyses. Even in well-designed and well-implemented studies, there are many sources of missing data, such as nonresponse in survey studies, or attrition in longitudinal studies. Beyond a negligible amount, missing data can reduce the representativeness and distort statistical inferences drawn from the sample [1,2]. Depending on the underlying mechanism of missingness, restricting data to complete cases can potentially lead to biased results and incorrect conclusions [1,3]. As a result, many approaches of handling missing data, such as maximum-likelihood estimation and imputations, are preferred to listwise deletion. Likelihood-based methods specify statistical models for all observed data and estimate the parameters of interest without discarding any partially observed cases [2,3,4,5]. In the presence of missing data, likelihood function-computes separate estimates for complete and incomplete variables and the likelihoods are maximized to produce an overall estimate [2,5]. Compared to imputations, maximum-likelihood (ML) algorithms often involve less preparation and have more inherent consistency, but they are limited to linear or log-linear models [4,5]. In addition to likelihood methods, imputation has been considered an effective method for managing item nonresponse bias by replacing missing data with plausible values and preserving available data and statistical power [3,6].

Imputation performance depends on the underlying causes of missing data [3]. More specifically, it depends on whether the data is intentionally or unintentionally missing [3]. Missing patterns can be primarily categorized into Missing Completely at Random (MCAR), Missing at Random (MAR), and Missing Not at Random (MNAR) [1,3]. MCAR occurs when missingness is unrelated to the data. Mathematically, it means that the probability of missing is equal to the overall missing rate [3]. In this case, a complete case analysis may produce unbiased estimates; however, this may be an unrealistic scenario and a difficult assumption to justify for most studies. MAR occurs when missingness depends on some observable covariate variables and MNAR occurs when missingness depends on unobserved factors, including the incomplete variable itself [3]. When the mechanism of missingness cannot be entirely accounted for by the observed covariates, it is considered to be non-ignorable [3]. Determining whether data is MAR or MNAR generally requires understanding or making assumptions about the process that led to values being missing.

There are many ways to impute missing data. Many methods are based on finding suitable candidates while maintaining the plausibility and semantic consistency in both statistical and machine-learning-based frameworks [7,8,9,10,11,12]. These methods may also perform differently under various conditions (e.g., missing pattern, missing rate, variable types, presence of functional dependencies among variables, etc.) [6,13,14]. Multiple imputation using the Chained Equation (MICE) is a widely used method and is available in several statistical software, such as R and SAS. The algorithm produces multiple complete datasets each imputed with plausible values that are selected from distributions specifically modeled for each incomplete variable [3,15]. For each imputed dataset, an estimate for the parameter of interest is generated and then pooled into one final estimate [3,15]. Another method of interest is fractional imputation. It identifies complete data as donors and missing values as recipients and creates imputation cells in which a set of donors is matched with each recipient based on the probability proportional to their fractional weights in a way that preserves the homogeneity of the data within each imputation cell [16]. The missing values are then jointly imputed by the assigned observed donor values within the cell [16]. A third method for consideration is generalized efficient regression-based imputation with latent processes (GERBIL). It utilizes a newly improved joint modelling (JM) approach, which historically did not efficiently accommodate data with the general structure (i.e., dataset with mixed data types, such as continuous, categorical, binary, and ordinal variables) [17,18]. It overcomes this drawback by sampling imputations from a latent joint multivariate normal distribution and has been shown to perform comparably to fully conditional specification (FCS) methods, such as MICE [17,18].

Multivariate missing data present a challenge for imputation methods due to interdependent incomplete variables and that each incomplete variable has its own set of correlated predictors. Having different incomplete variables to be imputed with a common set of correlated covariates may affect the efficacy of imputation. Similarly, we do not know how they perform towards multivariate missingness in real life. The goal of this study is to compare the three popular imputation methods: sequential multiple imputation (R package: MICE [15]), fractional hot-deck imputation (R package: FHDI [16]), and generalized efficient regression-based imputation with latent processes (R package: GERBIL [17]), under MAR and MNAR missing data assumptions using 2020 Behavioral Risk Factor Surveillance System (BRFSS) and 2017 to March 2020 National Health and Nutrition Examination Survey (NHANES) datasets. To the best of our knowledge, we are the first to compare those three methods by using public health data files and Monte Carlo simulation studies.

The rest of the paper is organized as following. Section 2 presents our proposed methods for comparison of the three imputation methods. Simulation results and study limitations are presented in Section 3. Section 4 contains the summary and conclusion.

## 2. Materials and Methods

Each full sample dataset was prepared by selecting variables of interests and related predictors, simplifying categories by regrouping as necessary, and omitting all the missing values from the original BRFSS and NHANES public datasets. Categorical variables to be imputed were limited to those having at least 20% occurrence in the full sample dataset. For BRFSS, we selected six diabetes-related variables to be made incomplete, among participants reported having diabetes. They correspond to age diagnosed with diabetes, number of feet checks by health professional in the past 12 months, number of A1c checks in the past 12 months, current insulin use, whether diabetes has affected eyes, and whether participant has ever taken a class on managing diabetes. Covariates are demographic variables that include state of residence, age groups, gender, race, education level, income level, marital status, employment status, health insurance status, BMI, exercise status, smoking status, alcohol status, general health status, time since last routine checkup, and household size.

For the NHANES dataset, we selected five variables related to chronic cardiovascular diseases to be made incomplete. They correspond to having high blood pressure, having high cholesterol, being overweight, high-sensitivity C-reactive protein (HsCRP) value, and whether participant was reducing salt intake. We included similar predictors as BRFSS, which were gender, age, race, education level, income level, marital status, health insurance status, access to healthcare, general health status, employment status, BMI, alcohol use, smoking status, and diet quality. To ensure effective imputation, we screened each set of predictors for significant associations with the outcome variables of interests.

Missing pattern for MAR data was generated via logistic function using significantly correlated predictors (Equation (1)). Categorical predictors were dummy-coded and continuous predictors were standardized. The probability of missing (*P*) was modeled after the binomial distribution with overall missing rate around 40% for each incomplete variable. If we let $x_{1}$ to $x_{n}$ be the set of correlated predictors, then *P* was calculated using
(1)$$P=\frac{\exp\left({\varphi_{0}+\varphi_{1}x}_{1}+\varphi_{2}x_{2}+\ldots +{\varphi_{n}x}_{n}\right)}{1+\exp\left(\varphi_{0}+{\varphi_{1}x}_{1}+\varphi_{2}x_{2}+\ldots +{\varphi_{n}x}_{n}\right)}$$
with $\varphi_{1}$ to $\varphi_{n}$ equal to −1 or 1 and varying values of $\varphi_{0}$ such that *P* has a mean of 0.4. Missing pattern for MNAR data was created similarly, but the probability of missing depended on both correlated predictors and the incomplete variable itself. For example, in the logistic function that generates missingness for the variable “age diagnosed with diabetes”, $x_{1}$ to $x_{n-1}$ would be correlated predictors and $x_{n}$ would be the variable “age diagnosed with diabetes” itself. For the purpose of comparing the imputation techniques under different magnitude of MNAR, we created two MNAR missing patterns where the outcome itself had a larger or smaller effect on the probability of missing. For small and large MNAR, $\varphi_{n}$ would be 0.1 and 3, respectively. Finally, we applied these missing patterns to the full sample dataset to generate total of 3 datasets with missing values under the MAR and MNAR (e.g., Large MNAR, Small MNAR) assumptions.

Descriptive statistics and logistic regression coefficients were calculated from each imputation method and compared to the corresponding “theoretical” (full sample) values. This process was then repeated 1000 times to account for the variability in the missing data generation step and the imputation process and to adequately compare the simulated results. For each outcome variable, we compare the results from complete-cases only, MICE, FHDI, and GERBIL in terms of the following: bias, relative bias, standard error, relative standard error, root mean square error, and relative root mean square error. For each missing pattern, 1000 simulated estimates ($\overset{-}{Q}$) from complete-case only and each imputation methods were averaged to obtain the final estimates ($E(\overset{-}{Q})$). Bias (*B*) was calculated by subtracting the full sample estimate (Q) from each final estimate ($B=E(\overset{-}{Q})-Q$). Standard errors (*SE*) were calculated by taking the square root of the variances among the 1000 estimates and root mean square errors (*RMSE*) were calculated using $\sqrt{B^{2}+{SE}^{2}}$, which present a compromise between bias and standard errors [3]. Relative measures were calculated by dividing their respective measures by the full sample value Q.

Variables representing number of feet checks by health professional in the past 12 months and number of A1c checks in the past 12 months were semicontinuous and very positively skewed. To account for this skewness in their density curves, we included a 2-step procedure in the MICE and GERBIL method for the MNAR datasets in addition to the standard imputation. This procedure consists of using binary indicators to pair with these semicontinuous variables during imputation and specify a zero or none-zero value for the imputed variables. This allows us to use a combination of binary and continuous imputation techniques in our efforts to produce less biased estimates for these 2 outcomes.

For the BRFSS dataset, we evaluated the coefficients of the logistic regression model using “current insulin use” as the outcome and the remainder of the incomplete variables “age diagnosed with diabetes”, “number of feet checks by health professional in the past 12 months”, “number of A1c checks in the past 12 months”, “whether diabetes has affected eyes”, and “whether participant has ever taken a class on managing diabetes” as well as age group, gender, health insurance status, BMI, and income level as predictors in the model. For the NHANES dataset, we selected “HsCRP value” as the outcome in the linear regression model with variables “having high blood pressure”, “having high cholesterol”, “being overweight”, and “reducing salt intake” as well as gender, age, health insurance status, general health status, access to healthcare, and BMI as the covariates in the model.

We used default MICE methods for binary (i.e., logistic regression) and continuous (i.e., predictive mean matching) variables. Number of iterations for MICE and GERBIL were 20 and 10, respectively. Descriptive statistics and regression coefficients from FHDI-imputed datasets were estimated by using fractional weights and default of 5 donors. Relative bias (*RB*), relative standard error (*RSE*), and relative root mean square error (*RRMSE*) are expressed as percentages. Simulated results were rounded to 4 decimal places.

## 3. Results

Simulation results are summarized in the order of increasing magnitude of non-ignorable missingness for comparison. Descriptive statistics and regression coefficients for selected variables from BRFSS and NHANES are presented in Table 1, Table 2, Table 3, Table 4, Table 5 and Table 6.

We used BRFSS and NHANES datasets for this study, because they are popular national public health datasets that are publicly available and therefore widely used for research purposes. These studies focusing on risk factors and disease associations have included similar variables to the ones we used here [19,20,21,22].

Expectedly, bias and RMSE increase as MNAR effects become larger for both complete-case and imputation. In missing data with predominantly MNAR patterns, imputation no longer produces reliable estimates [3,15,16,17]. Therefore, statistical inferences based on imputed data can lead to incorrect conclusions. However, in many cases, they have outperformed complete cases and significantly reduced bias and RMSE under the MNAR missing mechanism. In our study, MNAR was generated using a mixture of significantly correlated observed covariates and the incomplete variable itself. However, in real life data, MNAR can depend on unobserved variables and completely deviate from the MAR pattern. This scenario was not replicated in our study; however, in this case, the missingness can potentially be made closer to MAR by including more relevant covariates in the analyses [3].

In general, imputation with MICE and GERBIL resulted in less bias and RMSE compared to complete-case analyses. FHDI produced similar or larger biases compared to complete cases in several cases. This is likely due to the limitations of the current R package to adequately handle mixed types of incomplete data that we used for this study. The package does allow users to designate nominal categorical variables to be non-collapsible, but this often leads to problems involving cases of insufficient donors. Therefore, we did not perform any manual manipulation or rational discretization towards FHDI imputation due to variability and implementation difficulty associated with running 1000 simulations. As a result, an automatic cell-collapsing procedure may have impacted the quality of imputation if the joint distribution was inappropriately specified [16]. In data with completely arbitrary missingness, FHDI may yield the most optimal results out of the four options we considered in this study.

Each imputation method introduces uncertainty to the imputed value, which is desirable in preventing unrealistically short confidence intervals and high rates of false positives [3]. While RMSE evaluates the method of both accuracy and precision, the lowest RMSE does not signal the best method [3]. Among the three imputation methods, GERBIL consistently produced the largest standard errors.

To mimic real life analyses, we included incomplete variables as predictors in our regression models, which has led to the significant exclusion of observations in complete-cases analyses and severely biased results in many cases. However, imputation did not always result in lower bias, standard error, or RMSE, compared to complete cases. Imputation can be more effective by using only strongly associated observed covariates for each incomplete variable; however, for multivariate missing, this was not achieved as each incomplete variable may be significantly associated with different covariates. Imputation may perform differently with varying degree of missingness, number of incomplete variables, and for rare events [13]. Here, we only considered limited simulation scenarios with a limited selection of study variables from BRFSS and NHANES datasets.

Many similar studies have compared methods available within the MICE R package, while some compared MICE to methods similar to FHDI, such as k-nearest neighbor (knn), and GERBIL, such as joint multivariate normal imputation (JM-MVN) [23,24,25,26]. In addition, some studies have compared statistical methods, such as MICE, to machine-learning methods [13,14,27]. We share similar limitations with these studies in that our results are generalizable to datasets with similar conditions to the ones we specified here.

## 4. Conclusions

Imputation is a resourceful technique for handling problems related to missing data. For regression analyses with both incomplete predictors and outcomes, imputations preserve a considerable amount of observed data and often produce more valid estimates than listwise deletion in multivariate missing data with MAR or MNAR patterns. Similarly, descriptive statistics can be better estimated for incomplete variables of interest if relevant covariates are collected and without substantial missing values. However, this process requires screening for significantly correlated covariates and other preparatory work. In real world analyses, we do not know what the parameter estimate would be if the data were not missing and therefore we do not know how our choice of imputation method performs. Using real public health data, we presented the Monte Carlo simulation results for three methods that utilize different underlying algorithms to replace missing data with plausible values in order to improve the accuracy and precision of the sample estimate.

MICE offers various methods of performing multiple imputation on each variable and does not require a suitable underpinning multivariate distribution to exist. However, the disadvantages of this fully conditional specification (FCS) method include the possibility of incorporating incoherent or mis-specified joint distribution and non-convergence across iterations [3,16,17,28]. On the other hand, GERBIL avoids this problem by adopting the joint modelling (JM) approach that has been further designed to be compatible with the general data structure [17]. It is also more computationally efficient than MICE in high dimensional data [17]. To run ten rounds of simulation, MICE (five iterations, twenty multiply-imputed datasets), FHDI (no non-collapsible categorical variables specified, no variance estimation), and GERBIL (twenty-five iterations, ten multiply-imputed datasets) used 8.68, 1.07, and 2.64 min, respectively. Lastly, FHDI presents a nonparametric approach to handle data with arbitrary missing patterns by appropriately matching observed values to missing and imputing by probability based on fractional weights [16]. However, it may not always be appropriate for mixed-data types. Unordered categorical variables may need manual adjustments to prevent issues related to insufficient donors; therefore, this method may involve additional specifications and trial and error [16].

Additionally, there are many other methods available, including those that are based on machine-learning algorithms [29,30]. Our suggestions for future research include evaluating ways of estimating variance after imputation, comparing various machine-learning-based methods of imputation, and identifying appropriate methods for imputing data with outliers.

## Figures and Tables

**Table 1 ijerph-20-01524-t001:** Summary of Simulation Results for Mean of BRFSS Variable “Age when Diagnosed with Diabetes”.

**Missing Mechanism: MAR**						
Imputation Method	*B*	*RB* (%)	*SE*	*RSE* (%)	*RMSE*	*RRMSE* (%)
Complete-case	0.7191	1.5332	0.0076	0.0162	0.7191	1.5332
MICE: pmm	−0.2458	0.5242	0.0131	0.0279	0.2462	0.5249
FHDI	−0.9888	2.1082	0.0072	0.0154	0.9888	2.1083
GERBIL	−0.353	0.7527	0.0536	0.1143	0.3571	0.7614
**Missing Mechanism: Small MNAR**						
Imputation Method	*B*	*RB* (%)	*SE*	*RSE* (%)	*RMSE*	*RRMSE*
Complete-case	1.2694	2.7066	0.007	0.0149	1.2694	2.7066
MICE: pmm	0.1534	0.3271	0.008	0.0171	0.1536	0.3275
FHDI	−1.0655	2.2718	0.0218	0.0465	1.0657	2.2722
GERBIL	0.052	0.1108	0.0528	0.1126	0.0741	0.1580
**Missing Mechanism: Large MNAR**						
Imputation Method	*B*	*RB* (%)	*SE*	*RSE* (%)	*RMSE*	*RRMSE*
Complete-case	8.2251	17.5373	0.0161	0.0343	8.2252	17.5374
MICE: pmm	5.192	11.07	0.0121	0.0258	5.192	11.0701
FHDI	7.0293	14.9875	0.0099	0.0211	7.0293	14.9875
GERBIL	5.4701	11.6631	0.045	0.0959	5.4703	11.6635

*B* = bias; *RB* = relative bias; *SE* = standard error; *RSE* = relative standard error; *RMSE* = root mean square error; *RRMSE* = relative root mean square error; pmm = predictive mean matching.

**Table 2 ijerph-20-01524-t002:** Summary of Simulation Results for Mean of BRFSS Variable “Number of Feet Checks by Health Professional in the Past 12 Months”.

**Missing Mechanism: MAR**						
Imputation Method	*B*	*RB* (%)	*SE*	*RSE* (%)	*RMSE*	*RRMSE* (%)
Complete-case	0.1366	5.7022	0.002	0.0835	0.1366	5.7012
MICE: pmm	0.1013	4.2285	0.0003	0.0125	0.1013	4.2279
FHDI	0.1415	5.9063	0.0076	0.3172	0.1417	5.9140
GERBIL	0.115	4.7998	0.0208	0.8681	0.1169	4.8790
**Missing Mechanism: Small MNAR**						
Imputation Method	*B*	*RB* (%)	*SE*	*RSE* (%)	*RMSE*	*RRMSE* (%)
Complete-case	0.2539	10.5992	0.0016	0.0668	0.2539	10.5968
MICE: pmm	0.2297	9.5869	0.0018	0.0751	0.2297	9.5868
MICE: logreg + pmm	0.2175	9.0798	0.0018	0.0751	0.2175	9.0776
FHDI	0.2101	8.7674	0.0103	0.4299	0.2103	8.7771
GERBIL: 1-step	0.2199	9.1783	0.0219	0.9140	0.221	9.2237
GERBIL: 2-step	0.1084	4.5252	0.0203	0.8472	0.1103	4.6035
**Missing Mechanism: Large MNAR**						
Imputation Method	*B*	*RB* (%)	*SE*	*RSE* (%)	*RMSE*	*RRMSE* (%)
Complete-case	0.9304	38.8318	0.0004	0.0167	0.9304	38.8314
MICE: pmm	0.7432	31.0186	0.0002	0.0083	0.7432	31.0184
MICE: logreg + pmm	0.728	30.3866	0.0002	0.0083	0.728	30.3840
FHDI	0.5941	24.7963	0.0036	0.1503	0.5941	24.7955
GERBIL: 1-step	0.8023	33.487	0.0212	0.8848	0.8026	33.4975
GERBIL: 2-step	0.6896	28.7811	0.0202	0.8431	0.6899	28.7938

*B* = bias; *RB* = relative bias; *SE* = standard error; *RSE* = relative standard error; *RMSE* = root mean square error; RRMSE = relative root mean square error; pmm = predictive mean matching; logreg = logistic regression.

**Table 3 ijerph-20-01524-t003:** Summary of Simulation Results for Mean of NHANES Variable “HsCRP”.

**Missing Mechanism: MAR**						
Imputation Method	*B*	*RB* (%)	*SE*	*RSE* (%)	*RMSE*	*RRMSE* (%)
Complete-case	0.5803	15.1983	0.0039	0.1021	0.5803	15.1979
MICE: pmm	0.1707	4.4694	0.0008	0.0210	0.1707	4.4706
FHDI	−0.1661	4.3493	0.0028	0.0733	0.1661	4.3501
GERBIL	−0.0723	1.8937	0.0222	0.5814	0.0756	1.9799
**Missing Mechanism: Small MNAR**						
Imputation Method	*B*	*RB* (%)	*SE*	*RSE* (%)	*RMSE*	*RRMSE* (%)
Complete-case	0.6638	17.3859	0.0041	0.1074	0.6639	17.3873
MICE: pmm	0.1942	5.087	0.0031	0.0812	0.1943	5.0887
FHDI	−0.1376	3.6025	0.0085	0.2226	0.1378	3.6089
GERBIL	0.0009	0.0246	0.0238	0.6233	0.0239	0.6259
**Missing Mechanism: Large MNAR**						
Imputation Method	*B*	*RB* (%)	*SE*	*RSE* (%)	*RMSE*	*RRMSE* (%)
Complete-case	1.655	43.3448	0	0.0000	1.655	43.3439
MICE: pmm	1.382	36.1944	0.0026	0.0681	1.382	36.1941
FHDI	1.0198	26.7078	0.0094	0.2462	1.0198	26.7082
GERBIL	1.099	28.7826	0.0328	0.8590	1.0995	28.7955

*B* = bias; *RB* = relative bias; *SE* = standard error; *RSE* = relative standard error; *RMSE* = root mean square error; *RRMSE* = relative root mean square error; pmm = predictive mean matching.

**Table 4 ijerph-20-01524-t004:** Summary of Simulation Results for Proportion of NHANES Variable “Having High Blood Pressure”.

**Missing Mechanism: MAR**						
Imputation Method	*B*	*RB* (%)	*SE*	*RSE* (%)	*RMSE*	*RRMSE* (%)
Complete-case	−0.0699	17.3139	0.0002	0.0495	0.0699	17.3114
MICE: logreg	0.0083	2.0617	0.0004	0.0991	0.0083	2.0556
FHDI	0.0623	15.4393	0.0021	0.5201	0.0624	15.4539
GERBIL	0.0051	1.2713	0.0032	0.7925	0.006	1.4860
**Missing Mechanism: Small MNAR**						
Imputation Method	*B*	*RB* (%)	*SE*	*RSE* (%)	*RMSE*	*RRMSE* (%)
Complete-case	−0.0753	18.6511	0.0002	0.0495	0.0753	18.6487
MICE: logreg	0.002	0.5003	0.0003	0.0743	0.002	0.4953
FHDI	0.0112	2.7679	0.0004	0.0991	0.0112	2.7738
GERBIL	−0.0054	1.3342	0.0033	0.8173	0.0063	1.5603
**Missing Mechanism: Large MNAR**						
Imputation Method	*B*	*RB* (%)	*SE*	*RSE* (%)	*RMSE*	*RRMSE* (%)
Complete-case	−0.2603	64.474	0.0004	0.0991	0.2603	64.4657
MICE: logreg	−0.2229	55.2122	0.0005	0.1238	0.2229	55.2032
FHDI	−0.2499	61.8784	0.0003	0.0743	0.2499	61.8900
GERBIL	−0.2354	58.2945	0.003	0.7430	0.2354	58.2989

*B* = bias; *RB* = relative bias; *SE* = standard error; *RSE* = relative standard error; *RMSE* = root mean square error; *RRMSE* = relative root mean square error; logreg = logistic regression.

**Table 5 ijerph-20-01524-t005:** Summary of Simulation Results for Regression Coefficient for BRFSS Variable “Whether Diabetes has Affected Eyes”.

**Missing Mechanism: MAR**						
Imputation Method	*B*	*RB* (%)	*SE*	*RSE* (%)	*RMSE*	*RRMSE* (%)
Complete-case	−0.4639	59.406	0.0046	0.5890	0.464	59.4110
MICE	−0.2066	26.4567	0.0063	0.8067	0.2067	26.4661
FHDI	−0.3933	50.3674	0.0001	0.0128	0.3933	50.3585
GERBIL	−0.1727	22.1189	0.0303	3.8796	0.1754	22.4584
**Missing Mechanism: Small MNAR**						
Imputation Method	*B*	*RB* (%)	*SE*	*RSE* (%)	*RMSE*	*RRMSE* (%)
Complete-case	−0.0901	11.5387	0.0097	1.2420	0.0906	11.6005
MICE	0.1092	13.9853	0.0006	0.0768	0.1092	13.9821
FHDI	−0.2222	28.4482	0.0048	0.6146	0.2222	28.4507
GERBIL	0.0255	3.2647	0.0342	4.3790	0.0426	5.4545
**Missing Mechanism: Large MNAR**						
Imputation Method	*B*	*RB* (%)	*SE*	*RSE* (%)	*RMSE*	*RRMSE* (%)
Complete-case	0.7235	92.6478	0.0158	2.0230	0.7237	92.6633
MICE	0.3396	43.4898	0.0112	1.4341	0.3398	43.5083
FHDI	−0.0956	12.2474	0.0021	0.2689	0.0957	12.2535
GERBIL	0.0399	5.1105	0.0669	8.5659	0.0779	9.9744

*B* = bias; *RB* = relative Bias; *SE* = standard error; *RSE* = relative standard error; *RMSE* = root mean square error; *RRMSE* = relative root mean square error.

**Table 6 ijerph-20-01524-t006:** Summary of Simulation Results for Regression Coefficient for NHANES Variable “Having High Cholesterol”.

**Missing Mechanism: MAR**						
Imputation Method	*B*	*RB* (%)	*SE*	*RSE* (%)	*RMSE*	*RRMSE* (%)
Complete-case	1.8683	347.4789	0.1076	20.0118	1.8714	348.0487
MICE	0.0738	13.7336	0.0002	0.0372	0.0738	13.7255
FHDI	0.2546	47.3445	0.0227	4.2218	0.2556	47.5373
GERBIL	0.0799	14.8515	0.1021	18.9889	0.1296	24.1034
**Missing Mechanism: Small MNAR**						
Imputation Method	*B*	*RB* (%)	*SE*	*RSE* (%)	*RMSE*	*RRMSE* (%)
Complete-case	1.2265	228.1141	0.037	6.8814	1.2271	228.2198
MICE	0.4352	80.9333	0.0107	1.9900	0.4353	80.9584
FHDI	−0.2359	43.8748	0.0031	0.5765	0.2359	43.8734
GERBIL	0.2013	37.4425	0.1089	20.2536	0.2289	42.5715
**Missing Mechanism: Large MNAR**						
Imputation Method	*B*	*RB* (%)	*SE*	*RSE* (%)	*RMSE*	*RRMSE* (%)
Complete-case	−1.3693	254.6583	0.0366	6.8070	1.3697	254.7410
MICE	−0.4281	79.617	0.0182	3.3849	0.4285	79.6937
FHDI	−0.5571	103.6038	0.0017	0.3162	0.5571	103.6112
GERBIL	−0.8072	150.1331	0.1505	27.9904	0.8212	152.7293

*B* = bias; *RB* = relative bias; *SE* = standard error; *RSE* = relative standard error; *RMSE* = root mean square error; *RRMSE* = relative root mean square error.

## Data Availability

We used the following publicly available data files: 2020 BRFSS data (https://www.cdc.gov/brfss/annual_data/annual_2020.html, accessed on 24 December 2022) and 2017–March 2020 (pre-pandemic) NHANES data (https://wwwn.cdc.gov/nchs/nhanes/continuousnhanes/default.aspx?cycle=2017-2020, accessed on 24 December 2022).

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
