# Peer review of "Empirical Comparison of Imputation Methods for Multivariate Missing Data in Public Health"

_ijerph, 2023, doi:10.3390/ijerph20021524_

Round 1
Reviewer 1 Report
The authors have compared the performance of three commonly-used imputation methods for three data sets under different missing data patterns. Overall, the paper is very interesting in comparing different methods for multiple data sets with different missing data patterns. The manuscript is well organized except for (1) missing enough literature in imputation methods and (2) literature dealing data without imputation, (3) mathematical formulae used for different scenarios in simulations, and (4) verification/motivation of the use of the three data sets.
Comments:
1. Although the author compared three commonly-used imputation methods, there are many other imputation methods in imputation. A more systematic presentation of missing data handling literature is needed. In addition to missing data imputation, an alternative way, the methods of directly using likelihood without imputing missing data are also important whereas not well presented in the manuscript. Please polish and rewrite the literature of (1) missing data imputation, and (2) using likelihood-based approach without imputation to deal with data with missing values.
2. The author needs to provide motivation and verification of using the three data sets for empirical evaluation of methods. Are three data sets unique and possess common characteristics of data in health statistics or biostatistics? Are these data sets well used by researchers in the literature? Please provide some important papers using these data sets.
3. In describing the missing data pattern for the three data sets, the authors are at high level without providing sufficient details. I would like to know specific the mathematically formulae and parameter values used in different scenarios of missing data mechanism in simulation. For each formula, please specify whether or not it belongs to MCAR, MAR and MNAR. In addition, are there missing values already in the "original" data sets, or the original data sets do not have missing values whereas the simulation masks some entries in data matrix based on missing data pattern scenarios?
Reviewer 2 Report
This article proposes an empirical comparison among three popular imputation methods (sequential multiple imputation, fractional hot-deck imputation, and generalized efficient regression-based imputation with latent processes) performed considering two real-world health datasets. The comparison was carried out by considering three different mechanisms of missingness: MAR, Small MNAR, and Large MNAR. The authors discuss the effects of the aforementioned methods in reducing bias.
In general, the article is well-written, the experimental setup is clearly described and the proposed comparison is interesting. However, I believe that the manuscript could
benefit from the following improvements:
- Although the manuscript is strictly focused on three imputation methods, I suggest providing an overview of other imputation approaches, such as:
https://doi.org/10.1109/TKDE.2018.2883103, https://doi.org/10.1109/ICDE.2013.6544847, https://doi.org/10.5441/002/edbt.2022.05, https://doi.org/10.1145/3394486.3403096
https://doi.org/10.1016/j.knosys.2021.107114, and https://doi.org/10.48550/arXiv.1702.00820.
-I also suggest adding a related work section in which the authors should discuss other similar comparison studies, their limitations, and in what aspects their
work differs from these ones.
-The metrics employed in the evaluation (such as RB, RSE and so on) should be briefly described.
-It would be interesting if the authors could expose some considerations on the execution time employed by the different algorithms in the Results section.
Round 2
Reviewer 2 Report
In this revised version, the paper has been improved substantially, and the authors solved all of the remarks. They discussed more in-depth experimental results and the evaluation metrics used in their tests. Moreover, a deeper presentation of the related literature has been added.